# S417 in the CC3 region of STIM1 is critical for STIM1-Orai1 binding and CRAC channel activation

Tao Yu[2], Xi Li[1], Qianqian Luo[1], Huajing Liu[1], Jing Jin[1], Shengjie Li[1], Jun He[1]

Store-operated Ca$^{2+}$ entry (SOCE) is a universal Ca$^{2+}$ influx pathway that is important for the function of many cell types. SOCE is controlled by the interaction of the ER Ca$^{2+}$ sensor STIM1 with the plasma membrane Ca$^{2+}$ channel Orai1. S417 is located in the third coiled-coil (CC3) domain of the C-terminus of STIM1. We found that single-point mutation of this residue (S417G) abolished STIM1 C-terminus interactions with Orai1. Mutation of S417 also abolished CAD-Orai1 binding and Orai1 channel activation, eliminated STIM1 puncta formation, and co-localization with Orai1 and SOCE. 2-APB was found to restore the binding of the STIM1 C-terminus mutant (S417G) to Orai1 and dose-dependently activate Orai1 channel. Both CBD and NBD of Orai1 are required for 2-APB–induced coupling between the Orai1 and STIM1 C-terminus mutant (S417G) and CRAC channel activation. We also demonstrated that 2-APB led to delayed activation of Orai1-K85E channel, although Orai1-K85E obviously impairs 2-APB–induced STIM1 C-terminus mutant (S417G)–Orai1 coupling. Our results suggest S417 in the CC3 domain of STIM1 is essential for STIM1–Orai1 binding and CRAC channel activation.

## Introduction

Store-operated Ca$^{2+}$ entry (SOCE) is the major route of Ca$^{2+}$ entry in both excitable and especially non-excitable cells and plays important roles in the control of gene expression, cell growth and differentiation, motility, secretion, tissue and organ development, and the immune response (Parekh & Putney, 2005; Feske, 2010; Prakriya & Lewis, 2015; Shim et al, 2015). Abnormal SOCE has been associated with different human disorders. Loss-of-function mutations in Orai1 and STIM1 genes cause CRAC channelopathies, involving immuno-deficiency and autoimmunity, muscular hypotonia, ectodermal dysplasia, and mydriasis (Feske et al, 2006; Picard et al, 2009; Lacruz & Feske, 2015). In contrast, STIM1 and Orai1 gain-of-function mutations were found in patients with tubular aggregate myopathy and Stormorken syndrome (STRMK) (Böhm et al, 2013; Misceo et al, 2014; Morin et al, 2014; Nesin et al, 2014; Morin et al, 2020). In addition, SOCE plays important roles in the pathophysiology of cardiovascular diseases, thrombus formation, tumor cell metastasis, and the pathogenesis of acute pancreatitis (Varga-Szabo et al, 2008; Braun et al, 2009; Yang et al, 2009; Zhu et al, 2018; Lee & Papachristou, 2019; Lu et al, 2022), etc.

Regulation of SOCE is a highly choreographed process that involves a complex conformational rearrangement of STIM1 proteins (Hogan et al, 2009; Shaw et al, 2013; Prakriya & Lewis, 2015). In resting cells replete with Ca$^{2+}$, STIM1 is distributed diffusely throughout the ER in an inactive state (Zhang et al, 2005; Baba et al, 2006; Wu et al, 2014). After ER Ca$^{2+}$ depletion, Ca$^{2+}$ releases from the luminal Ca2+-binding EF hand, leading to the unfolding of the EF-sterile α motif (SAM) domain, and the conformational extension of the cytoplasmic STIM1 C-terminus (STIM1-CT) (Korzeniowski et al, 2010; Muik et al, 2011; Yu et al, 2013; Zhou et al, 2013). STIM1-CT contains three coiled-coil domains: CC1 (residues 238–343), CC2 (residues 363–389), and CC3 (residues 399–423). CC2 and CC3 are located in a ~100-aa region in STIM1-CT, variously termed the CRAC activation domain (CAD; residues 342–448), or STIM1-Orai1 activating region (SOAR; residues 344–442) or coiled-coil domain b9 (CCb9, aa 339–446) (Kawasaki et al, 2009; Park et al, 2009; Yuan et al, 2009; Soboloff et al, 2012). Conformational extension of STIM1-CT exposes its polybasic domain at the distal end (Korzeniowski et al, 2010; Muik et al, 2011; Yu et al, 2013; Zhou et al, 2013). This polybasic domain facilitates the recruitment of STIM1 oligomers to ER–PM junctions by interacting with acidic phospholipids in the PM, where Orai1 accumulates in the areas of plasma membrane–apposed STIM1 puncta, and the CAD or SOAR segment of STIM1 binds to C and N termini of Orai1 protein (Liou et al, 2007; Park et al, 2009; Yuan et al, 2009; Korzeniowski et al, 2010; Muik et al, 2011; Zhou et al, 2013), which is sufficient to activate CRAC channels and induce constitutive Ca$^{2+}$ influx (Kawasaki et al, 2009; Park et al, 2009; Yuan et al, 2009; Soboloff et al, 2012). The structurally best-defined region in CAD is CC2 that establishes the binding interactions with Orai1 (Stathopulos et al, 2013). Although recent studies have shown that a coiled-coil clamp involving the CC1 and CC3 domains is essential in controlling STIM1 activation, the role of CC3 remains to be elucidated (Muik et al, 2011; Fahrner et al, 2014; Maus et al, 2015; Rathner et al, 2021).

Here, we show that a single serine residue in CC3 segment is absolutely required for STIM1-CT binding to Orai1. Mutation of S417 in CC3 hinders STIM1 puncta formation and impairs CAD-Orai1

[1]Division of Histology and Embryology, School of Basic Medical Sciences, Tongji Medical College, Huazhong University of Science and Technology, Wuhan, China [2]Department of Clinical Laboratory, Wuhan Children's Hospital (Wuhan Maternal and Child Healthcare Hospital), Tongji Medical College, Huazhong University of Science and Technology, Wuhan, China

Correspondence: junhe@mails.tjmu.edu.cn

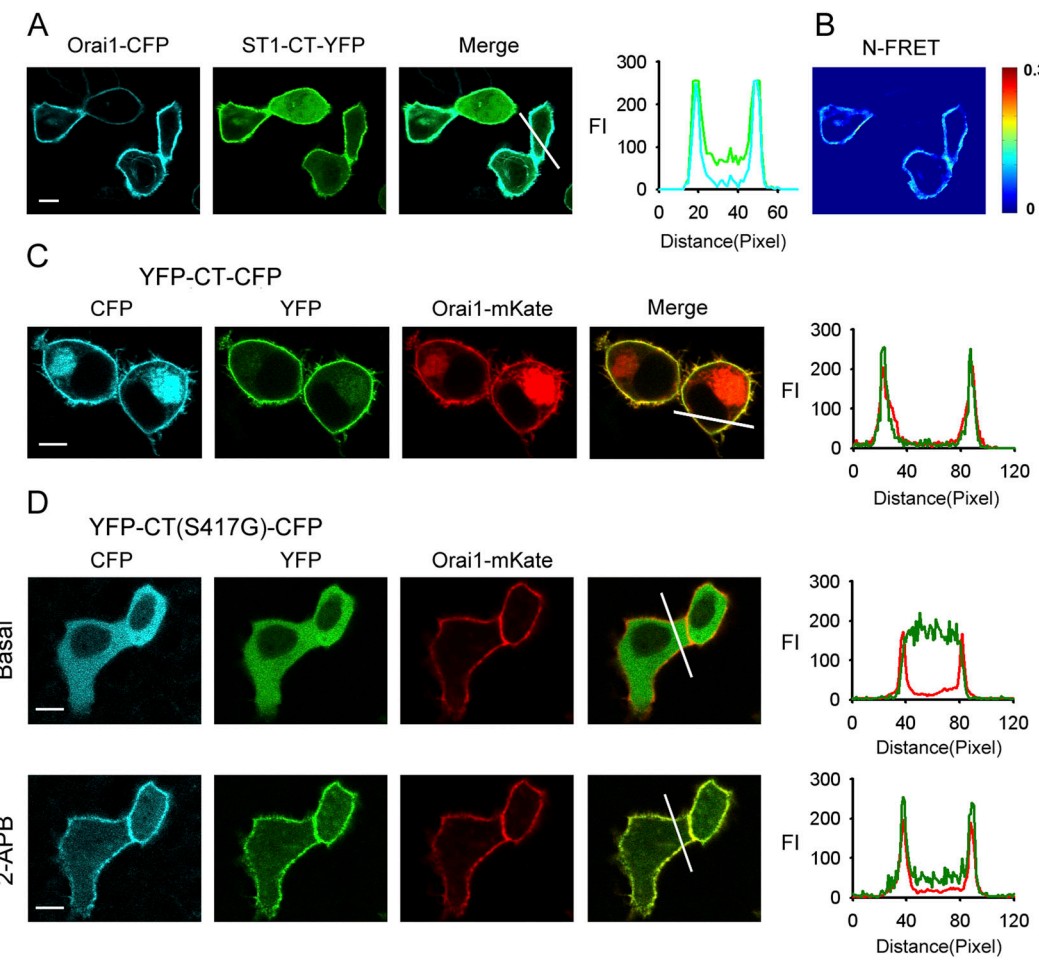

**Figure 1. S417 mutation abolishes STIM1 C-terminus–Orai1 interaction.**
**(A)** YFP-labeled whole STIM1 C-terminus (ST1-CT-YFP) coexpressed with Orai1-CFP in HeLa cells exhibited clear membrane targeting. Line-scan intensity plots show the distribution of ST1-CT-YFP (green lines) and Orai1-CFP (blue line). **(B)** N-FRET live cell images of HeLa cells coexpressing ST1-CT-YFP and Orai1-CFP. **(C)** Double-labeled whole STIM1 C-terminus (YFP-CT-CFP) expression together with Orai1-mKate also exhibited clear membrane targeting. Line-scan intensity plots show the distribution of YFP-CT-CFP (green lines) and Orai1-mKate (red line). **(D)** YFP-CT (S417G)-CFP coexpressed with Orai1-mKate in HeLa cells was redistributed from the cytoplasm to the PM upon application of 50 μM 2-APB. Line-scan intensity plots show the distribution of YFP-CT (S417G)-CFP (green lines) and Orai1-mKate (red line) before (upper panel) and 5 min after (lower panel) application of 50 μM 2-APB. Bar = 10 μm.

binding, thereby abolishing CRAC channel activation. 2-Amino-ethoxydiphenyl borate (2-APB), a popular pharmacological agent in the study of CRAC/store-operated channels, can restore the binding of the STIM1-CT mutant (S417G) to Orai1 and activate Orai1 channel in a dose-dependent manner. Our data indicated that S417 in CC3 domain of STIM1 might be a crucial element for STIM1 function, and activation of CRAC and CC3 domain of STIM1 plays an essential role for these processes. We also present a new finding about 2-APB's action on CRAC channel, which provides a solid base for better understanding of SOCE activation.

## Results

### Mutation of S417 abolishes STIM1 C-terminus interactions with Orai1

The YFP-labeled whole STIM1 C-terminus (233–685) (ST1-CT-YFP) expression together with Orai1-CFP in HeLa cells led to a clear

redistribution of ST1-CT-YFP with partial plasma membrane and cytosolic targeting (Fig 1A). We measured fluorescence resonance energy transfer (FRET) between Orai1-CFP and ST1-CT-YFP, which demonstrated the direct interaction between whole STIM1 C-terminus and Orai1 (Fig 1B). Confocal microscopy showed that the double-labeled whole STIM1 C-terminus (YFP-CT-CFP) coexpressed with Orai1-mKate was predominantly found at the PM, whereas STIM1 C-terminus mutant (YFP-CT (S417G)-CFP) expression together with Orai1-mKate was completely localized in the cytoplasm, indicating that mutation of S417 abolishes STIM1 C-terminus interactions with Orai1 (Fig 1C and D).

### 2-APB induces STIM1 C-terminus mutant (S417G) and Orai1 to undergo rapid reorganization into co-localized PM clusters

Here, in HeLa cells coexpressing STIM1 C-terminus mutant (YFP-CT (S417G)-CFP) and Orai1-mKate, we found that the Orai1-mKate protein was exclusively and relatively uniformly located in the

PM, and YFP-CT (S417G)-CFP was completely cytoplasmic before the addition of 2-APB (Fig 1D, upper). 30 s after application of 50 μM 2-APB, YFP-CT (S417G)-CFP redistributed to the PM, where it co-localized with Orai1-mKate, suggesting that the two proteins formed a complex (Fig 1D, bottom).

### The S417G OASF sensor mutant showed a significant enhancement of FRET

WT or mutant (S417G) YFP-CT-CFP (aa 233–685), YFP-OASF-CFP (aa 233–474), and YFP-CAD-CFP (aa 342–448) intramolecular FRET sensors were constructed to investigate the mechanism by which 2-APB restores the interaction between STIM1 C-terminus mutant (S417G) and Orai1. Each construct contained the CAD of STIM1, which is the key segment involved in this interaction (Fig 2A). The S417G OASF and S417G CT sensor mutants showed significant FRET enhancement compared with the OASF WT form, whereas the FRET of the S417G CAD sensor mutant is similar to that of the WT sensor (Fig 2B–E).

### S417 is required for STIM1-Orai1 binding and CRAC channel activation

Exogenous expression of the CAD/SOAR domain along with Orai1 has been shown to result in constitutive co-localization and binding of CAD/SOAR to Orai1 (Park et al, 2009; Yuan et al, 2009). Confocal microscopy imaging showed that YFP-CAD-WT was pre-dominantly found at the PM, whereas YFP-CAD (S417G) was com-pletely localized in the cytoplasm, indicating reduced binding of CAD (S417G) to Orai1 (Fig 3A and B). Accordingly, we observed the robust resting N-FRET between Orai1-CFP and YFP-CAD (Fig 3C). By contrast, N-FRET between Orai1-CFP and mutant YFP-CAD (S417G) was strongly reduced (Fig 3C).

To investigate whether the S417G mutation abolishes CRAC channel activation by CAD, we coexpressed Orai1 and WT or mutant CAD in HeLa cells. Although CAD-WT resulted in strong constitutive $Ca^{2+}$ influx, mutant CAD-S417G failed to induce $Ca^{2+}$ influx (Fig 3D–F).

A visible consequence of STIM1 oligomerization is the formation of STIM1 puncta at ER–PM junctions (Soboloff et al, 2012; Prakriya & Lewis, 2015). We analyzed the role of S417 in CC3 for STIM1 puncta formation by time-lapse confocal microscopy in cells expressing WT or mutant (S417G) mCherry-STIM1 together with Orai1-YFP. In nonstimulated cells, STIM1-WT localized to the bulk ER away from the PM, whereas Orai1 was distributed homogenously in the PM (Fig 4A). After ER store depletion with TG, STIM1-WT formed puncta, translocated to ER-PM junctions, and co-localized with Orai1 (Fig 4A). Likewise, mutant STIM1-S417G was distributed diffusely in ER in resting cells (Fig 4B). Surprisingly, upon store depletion, STIM1-S417G did not change its distribution and failed to redistribute into discrete puncta (Fig 4B). In addition, the distribution of Orai1-YFP remained homogenous after store depletion without signs of puncta formation and no significant co-localization with mCherry-STIM1-S417G (Fig 4B). Consequently, mCherry-STIM1-WT resulted in strong $Ca^{2+}$ influx; mCherry-STIM1- S417G failed to induce $Ca^{2+}$ influx (Fig 4C–E). To further investigate the role of S417 in STIM1 homo-merization, we measured relative FRET between STIM1 proteins in live cells. In cells expressing YFP-STIM1 and CFP-STIM1, we found a robust increase in the normalized IDA/IDD ratio after TG application

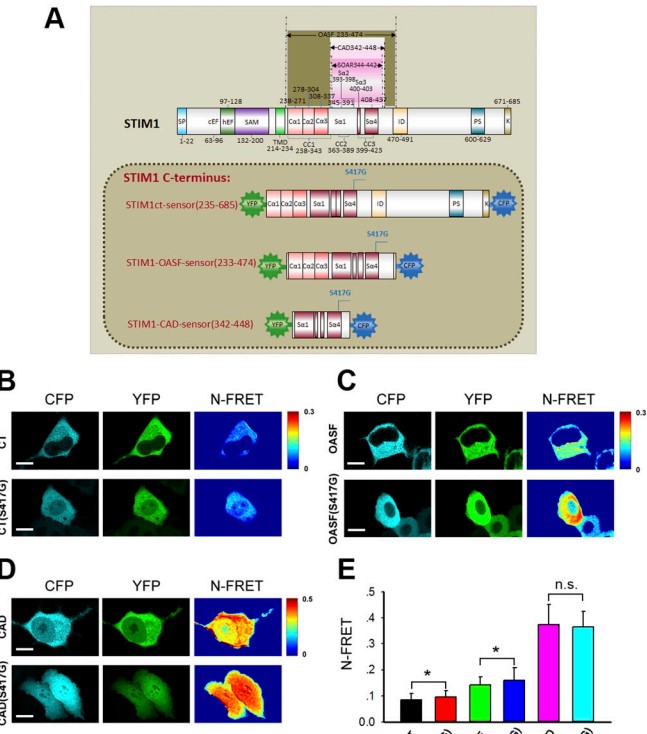

**Figure 2. S417G OASF sensor mutant showed a significantly higher FRET than WT OASF.**

**(A)** Schematic illustration of STIM1 protein and conformational sensors used in these studies. YFP-CT (S417G)-CFP contains the complete STIM1 C-terminus (S417G); YFP-OASF(S417G)-CFP is the OASF (S417G) region; and YFP-CAD (S417G)-CFP comprises the CAD (S417G) of STIM1. All proteins were labeled with YFP/CFP as shown. **(B, C, D)** Representative confocal microscopy images of YFP, CFP, and FRET signals from the WT or mutant (S417G) YFP-CT-CFP (aa 233–685), YFP-OASF-CFP (aa 233–474), and YFP-CAD-CFP (aa 342–448) intramolecular FRET sensors expressed in HeLa cells. **(E)** Block diagram summarizing NFRET of double-labeled STIM1 fragment mutants: YFP–CT–CFP (WT), YFP–CT (S417G)–CFP, YFP–OASF–CFP (WT), YFP–OASF (S417G)–CFP, YFP–CAD–CFP(WT), and YFP–CAD (S417G)–CFP (from left to right, n = 20, 31, 20, 32, 20, and 38). Graphs show mean ± SD. *$P < 0.05$; **$P < 0.001$. Bar = 10 μm.

compared with cells with filled $Ca^{2+}$ stores. By contrast, store de-pletion in cells expressing mutant YFP-STIM1-S417G and CFP-STIM1-S417G did not induce a significant FRET increase (Fig 4F and G). Together, these results indicated that mutation of S417 abolishes STIM1 function.

### 2-APB dose-dependently activated the Orai1 channel in store-replete HeLa cells coexpressing STIM1 C-terminus mutant (S417G) and Orai1

In our experiments, the red genetically encoded $Ca^{2+}$ indicator (GECI) CMV-R-GECO1.2 was chosen to examine the effect of different concentrations of 2-APB on cytosolic calcium in HeLa cells co-transfected with YFP-CT (S417G)-CFP, Orai1-CFP, and CMV-R-GECO1.2 (Fig 5A–C). As shown in Fig 5B, despite stores remaining full, coexpression of double-labeled STIM1 C-terminus mutant (YFP-CT (S417G)-CFP) caused massive increases in $Ca^{2+}$ entry, with the

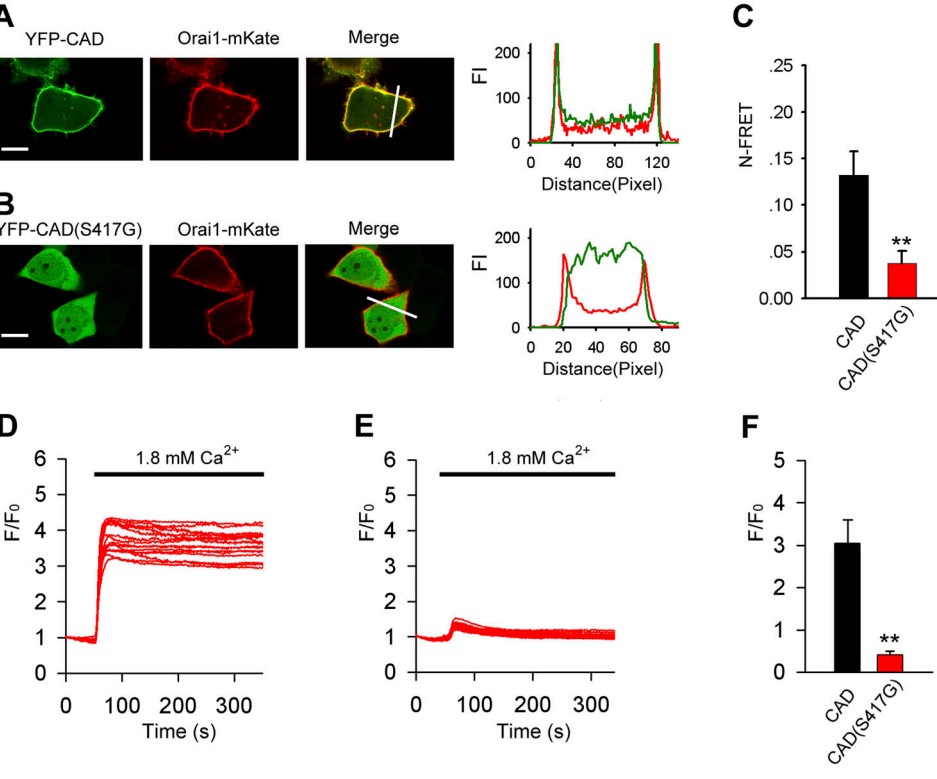

Figure 3. S417 mutation abolishes CAD–Orai1 interaction.
**(A)** Representative confocal images of HeLa cells coexpressing YFP-CAD and Orai1-mKate. Line-scan intensity plots show the distribution of YFP-CAD (green lines) and Orai1-mKate (red line). **(B)** Representative confocal images of cells coexpressing YFP-CAD (S417G) and Orai1-mKate. Line-scan intensity plots show the distribution of YFP-CAD (S417G) (green lines) and Orai1-mKate (red line). **(C)** Averaged N-FRET of Orai1-CFP and WT or mutant (S417G) YFP-CAD expressed in HeLa cells without TG stimulation in 0 mM $Ca^{2+}$ (from left to right, n = 55 and 55). **(D, E)** $Ca^{2+}$ influx in HeLa cells coexpressing Orai1 and WT-CAD (D) or mutant CAD-S417G (E). **(D, E, F)** Averaged peak $[Ca^{2+}]_i$ values associated with $Ca^{2+}$ influx in HeLa cells coexpressing Orai1 and WT-CAD or mutant CAD-S417G (from left to right, n = 32 and 32) shown in panels (D, E). Graphs show mean ± SD. *$P < 0.05$; **$P < 0.001$. Bar = 10 $\mu m$.

attainment of maximal $Ca^{2+}$ peak upon the addition of higher concentrations of 2-APB(>20 $\mu M$), followed by rapid inhibition. Instead, lower 2-APB concentrations (<5 $\mu M$) appeared to potentiate but not inhibit $Ca^{2+}$ influx. According to experiment A, representative fluorescence pseudo-color images of CMV-R-GECO1.2 in live cells before and after application of 50 $\mu M$ 2-APB were acquired at 0, 446, 500, and 600 s, respectively (Fig 5C). In addition, using the green genetically encoded ER $Ca^{2+}$ indicator ER-GCaMP6, we demonstrated that concentrations of 2-APB in the range of 1~100 $\mu M$ did not affect the release of $Ca^{2+}$ from intracellular stores in HeLa cells coexpressing YFP-CT (S417G)-CFP and Orai1-CFP.

### 2-APB induces the extension of STIM1 C-terminus mutant (S417G)

YFP-CT (S417G)-CFP conformational sensor enabled us to investigate the molecular mechanism by which 2-APB induces the activation of Orai1 through its interaction with the STIM1 C-terminus mutant (S417G). In HeLa cells transfected with YFP-CT (S417G)-CFP, a slight decrease in FRET was observed in the presence of 50 $\mu M$ 2-APB, suggesting rearrangement to an extended conformation (Fig S1A–C). The YFP-CT (S417G)-CFP sensor formed cytosolic aggregates but did not localize to the PM after 2-APB application (Fig S1D). Treatment with 5 $\mu M$ 2-APB resulted in a slight FRET decrease, suggesting that the STIM1 C-terminus mutant (S417G) assumed an extended conformation even at a low concentration of 2-APB (Fig S1E and F). YFP-CT (S417G)-CFP was still uniformly distributed in the cytoplasm when coexpressed with Orai1-mKate (Fig S2A); however, YFP-CT (S417G)-CFP was rapidly localized to the PM upon the addition of 50 $\mu M$ 2-APB, which was accompanied by a slight decrease

in FRET (Fig S2A–C); a similar but less pronounced effect was observed by treatment with 5 $\mu M$ 2-APB (Fig S2D–F). These data suggested that the slightly extended conformation of the STIM1 C-terminus mutant (S417G) induced by 2-APB may be a potential mechanism for STIM1 CT (S417G)–Orai1 binding and CRAC channel activation.

### The C- and N-terminal STIM1 binding sites on Orai1 are required for 2-APB–induced STIM1 C-terminus mutant (S417G)–Orai1 coupling

2-APB appeared to promote the interaction of the STIM1 C-terminus mutant (S417G) and Orai1. To identify the sites in Orai1 that were involved in this interaction, we generated truncation constructs of Orai1-mKate lacking the cytoplasmic C-terminus (Orai1-ΔC-mKate) and N-terminus (Orai1-ΔN-mKate). YFP-CT (S417G)-CFP overexpressed in HeLa cells redistributed from the cytoplasm to the PM where it co-localized with Orai1-mKate upon 2-APB application (Figs 1D and S2A); moreover, both proteins exhibited an increase in FRET after 2-APB application (Fig 7A and B), suggesting an direct interaction between them. However, cytoplasmic YFP-CT (S417G)-CFP failed to co-localize with overexpressed Orai1-ΔC-mKate (a.a. 1–272) in the presence of 2-APB (Fig 6A). Truncation of the cytoplasmic N-terminus of Orai (Orai1-ΔN-mKate; a.a. 90–301) also diminished the redistribution of YFP-CT (S417G)-CFP or its co-localization with Orai1-ΔN-mKate at the cell surface with 2-APB treatment (Fig 6B), implying that mutations at the N-terminal site significantly impair 2-APB–induced STIM1 C-terminus mutant (S417G)–Orai1 coupling, although less significantly than mutations at the C-terminal site. To identify the motif within Orai1 that mediates this interaction, we generated Orai1

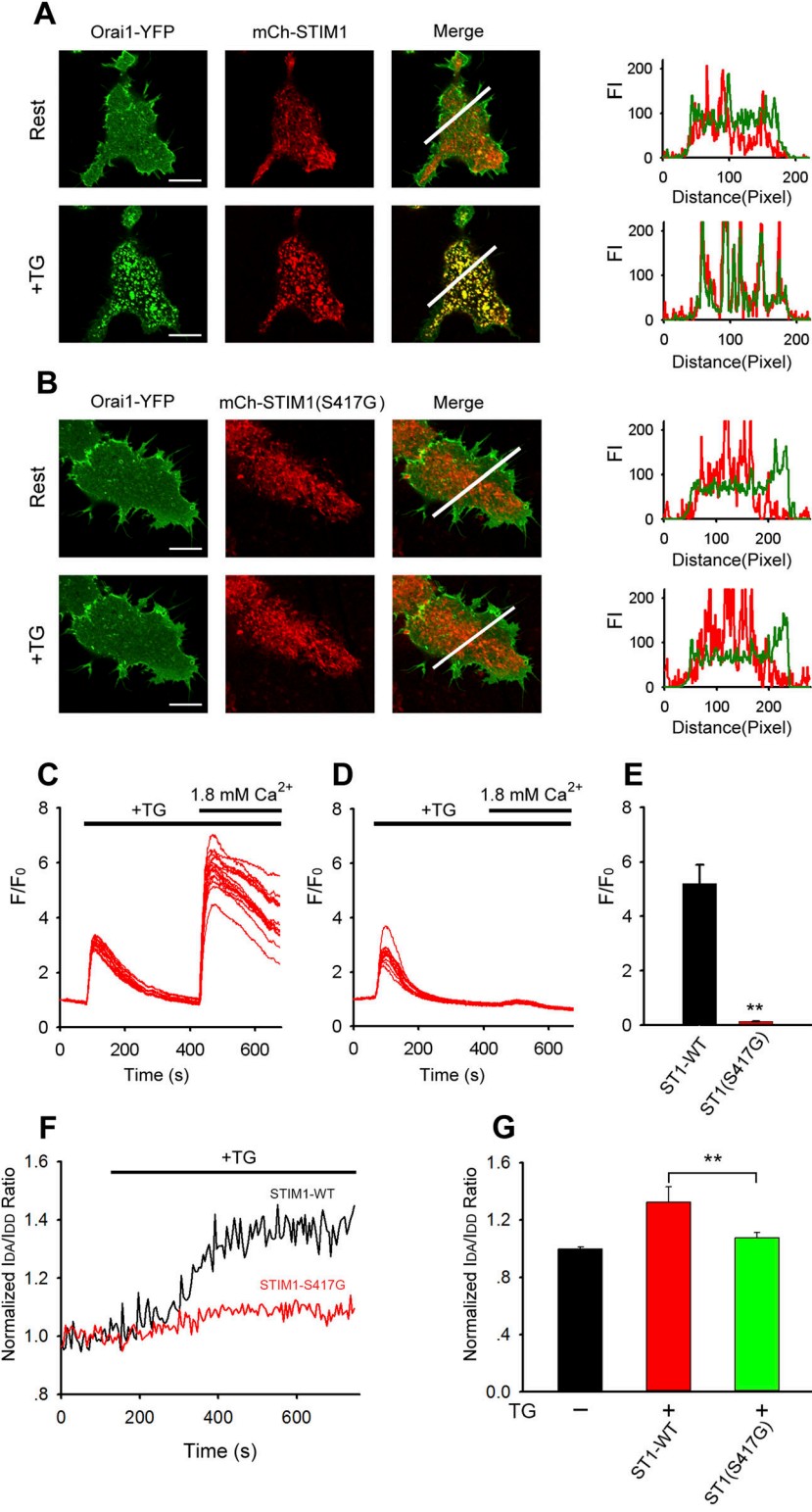

**Figure 4. S417 is essential for STIM1 puncta formation and co-localization with Orai1.**
**(A, B)** Representative confocal images of HeLa cells expressing Orai1-YFP and WT or mutant (S417G) mCherry-STIM1 before and after stimulation with 1 $\mu$M TG in $Ca^{2+}$-free Ringer's solution for 5 min. Line-scan intensity plots show the distribution of WT or mutant (S417G) mCherry-STIM1 (red lines) and Orai1-mKate (green line) before (upper panel) and 5 min after (lower panel) application of 1 $\mu$M TG. **(C, D)** TG-induced SOCE in HeLa cells coexpressing Orai1 and WT-STIM1 (C) or mutant STIM1-S417G (D). **(C, D, E)** Averaged peak $[Ca^{2+}]_i$ values associated with SOCE in HeLa cells coexpressing Orai1 and WT-STIM1 or mutant STIM1-S417G (from left to right, n = 29 and 26) shown in panels (C, D). **(F)** Representative traces showing the normalized $I_{DA}/I_{DD}$ ratio between WT or mutant (S417G) CFP-STIM1 and YFP-STIM1 (from left to right, n = 12, 12, and 12). **(G)** Averaged $I_{DA}/I_{DD}$ ratio before and after stimulation with 1 $\mu$M TG. Graphs show mean ± SD. *$P < 0.05$; **$P < 0.001$. Bar = 10 $\mu$m.

mutants with deletions in the CBD (Orai1-ΔCBD-mKate; Δa.a. 272–292) or NBD (Orai1-ΔNBD-mKate; Δa.a. 73–85). The confocal imaging and FRET experiments show that ΔCBD Orai1 mutations abolished STIM1 C-terminus mutant (S417G)–Orai1 association

induced by 2-APB and Orai1-ΔNBD and also significantly reduced these couplings (Figs 6C and D and 7C–F). These findings indicate that both CBD and NBD of Orai1 mediates 2-APB–induced STIM1 C-terminus mutant (S417G)–Orai1 association.

**Figure 5. 2-APB dose-dependently induced large Ca²⁺ entry in HeLa cells co-transfected with YFP-CT (S417G)-CFP and Orai1-CFP.**
HeLa cells were co-transfected with YFP-CT (S417G)-CFP, Orai1-CFP, and CMV-R-GECO1.2 constructs. After 48 h, live cells were examined in a confocal microscope. **(A)** Representative confocal images of HeLa cells coexpressing YFP-CT (S417G)-CFP, Orai1-CFP, and CMV-R-GECO1.2 before and after application of 50 μM 2-APB. **(B)** Different concentrations of 2-APB–induced Ca²⁺ influx reported by the red genetically encoded Ca²⁺ indicator CMV-R-GECO1.2. **(C)** Representative fluorescence pseudo-color images of CMV-R-GECO1.2 in live cells before and after application of 50 μM 2-APB were obtained at 0, 446, 500, and 600 s, respectively. Bar = 10 μm.

## Orai1 C-terminal residues L273 and L276 mediate coupling to STIM1 C-terminus mutant (S417G) induced by 2-APB

Previous studies identified L273 and L276 in the C-terminus of Orai1 as critical residues for interaction with STIM1. To assess the contribution of these residues to the STIM1 C-terminus mutant (S417G)–Orai1 coupling induced by 2-APB, we generated two Orai1 mutants (Orai1-L273D-mKate and Orai1-L276D-mKate). Cytoplasmic YFP-CT (S417G)-CFP failed to co-localize with coexpressed Orai1-L273D-mKate or Orai1-

L276D-mKate at the PM upon addition of 2-APB (Fig 8). These results indicate that Orai1 L273 and L276 play key roles in the interaction of STIM1 C-terminus mutant (S417G) and Orai1 induced by 2-APB.

## Orai1 CBD and NBD are required for 2-APB–induced STIM1-CT mutant (S417G)–Orai1 binding and CRAC channel activation

For functional analyses, the original set of Orai1 mutants was co-transfected with YFP-CT (S417G)-CFP in HeLa cells. Constitutive Ca²⁺

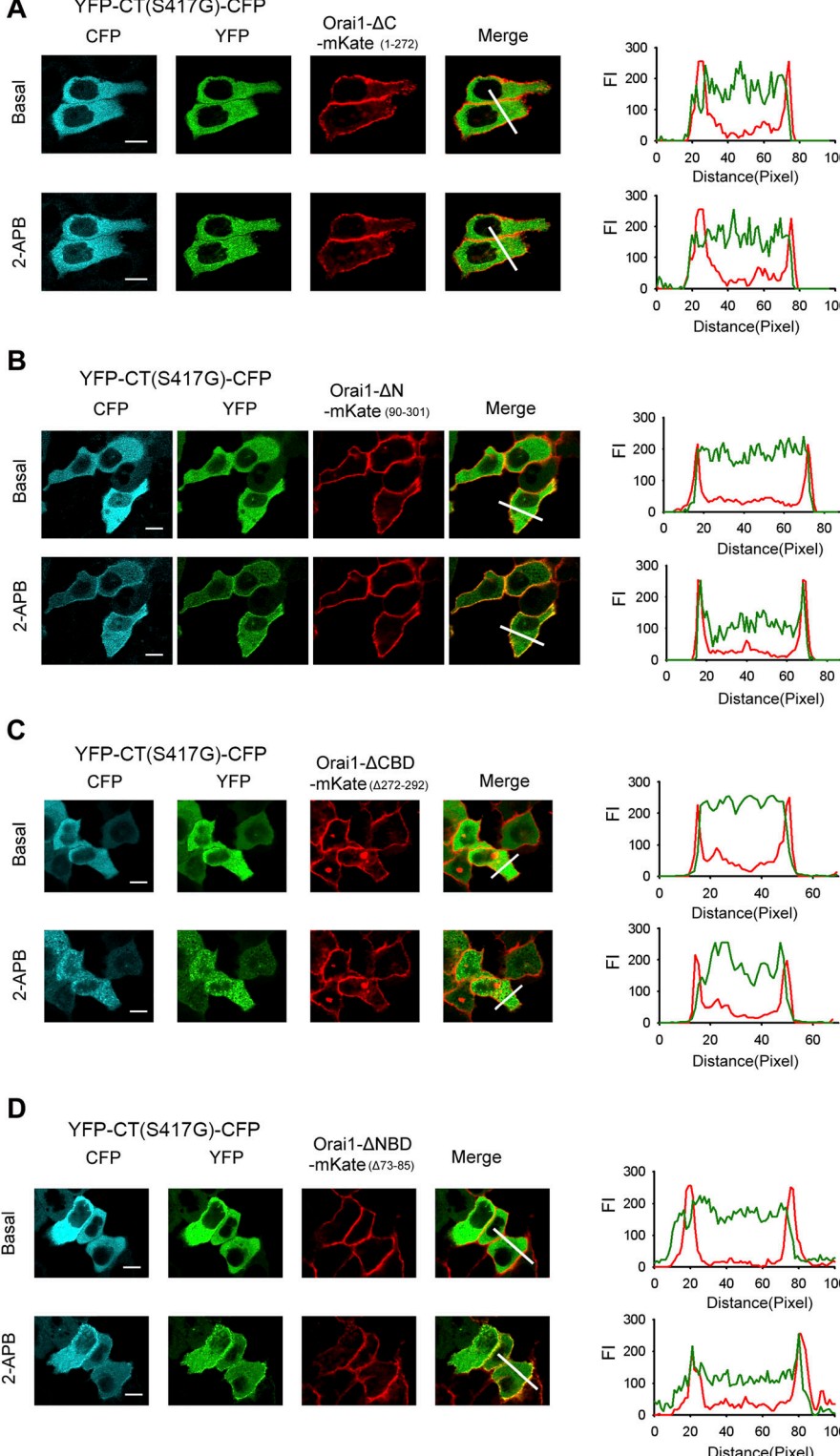

**Figure 6. Both CBD and NBD of Orai1 is required for 2-APB–induced coupling between the STIM1 C-terminus mutant (S417G) and Orai1.**

((A, B, C, D) left) Representative fluorescent images of the indicated YFP-CT (S417G)-CFP construct coexpressed with the indicated Orai1 mutants before and after application of 2-APB (50 $\mu$M). Note that when coexpressed with Orai1-ΔN-mKate or Orai1-ΔNBD-mKate but not with Orai1-ΔC-mKate or Orai1-ΔCBD-mKate, the cellular distribution of YFP-CT-CFP is changed after the addition of 2-APB. However, the redistribution of YFP-CT (S417G)-CFP coexpressed by these N-terminal mutants is weaker than coexpressed by WT Orai1. ((A, B, C, D) right) Line-scan intensity plots depicting the distribution of YFP-CT (S417G)-CFP (green line) and Orai1 mutants (red line) between the cytosol and PM, as indicated by the solid line before (upper panel) and 5 min after (lower panel) application of 2-APB. Bar = 10 $\mu$m.

entry in cells coexpressing YFP-CT (S417G)-CFP and Orai1-mKate was negligible; however, a massive increase was observed upon addition of 50 $\mu$M 2-APB despite Ca$^{2+}$ stores remaining full (Fig 9A). In contrast to observations by confocal imaging, mutations (L273D or L276D) or truncations at either terminus of Orai1 (ΔNBD or ΔCBD) abolished 2-APB–triggered channel activity (Fig 9B–F). These results indicate that both cytosolic portions of Orai1 are essential for 2-APB–mediated Orai1 activity.

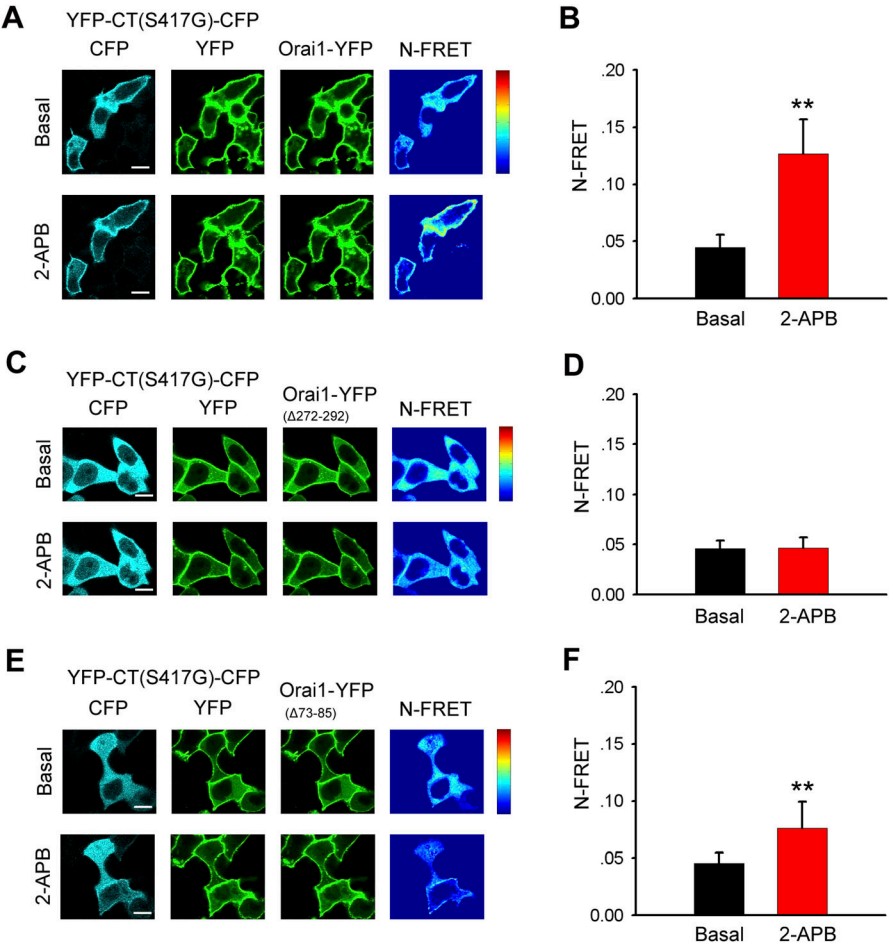

**Figure 7. Interaction between Orai1 and STIM1 C-terminus mutant (S417G) induced by 2-APB depends on the CBD and NBD of Orai1.**
**(A)** Localization and N-FRET live cell images of HeLa cells coexpressing YFP-CT (S417G)-CFP and Orai1-YFP. **(B)** Block diagram of calculated N-FRET in the PM (from left to right, n = 32 and 32). **(C)** Localization and N-FRET live cell images of cells coexpressing YFP-CT (S417G)-CFP and Orai1-ΔCBD-YFP. **(D)** Block diagram of calculated N-FRET for the PM (from left to right, n = 36 and 36). **(E)** Localization and N-FRET live cell images of cells coexpressing YFP-CT (S417G)-CFP and Orai1-ΔNBD-YFP. **(F)** Block diagram summarizing calculated N-FRET for the PM (from left to right, n = 36 and 36). Graphs show mean ± SD. *$P < 0.05$; **$P < 0.001$. Bar = 10 $\mu m$.

## 2-APB induced STIM1-CT mutant (S417G)–Orai1–K85E binding and Orai1-K85E channel activation

Previous studies identified K85 in the N-terminus of Orai1 as critical residues for store-operated gating of CRAC channels. To assess the contribution of this residue to the STIM1 C-terminus mutant (S417G)–Orai1 coupling induced by 2-APB, we generated Orai1 mutants (Orai1-K85E-CFP and Orai1-K85E-mKate). The single point mutation of the cytoplasmic N-terminus of Orai1 (Orai1-K85E) obviously diminished the redistribution of YFP-CT (S417G)-CFP or its co-localization with Orai1-K85E at the cell surface with 2-APB treatment (Fig 10A), implying that this mutation at the N-terminal site significantly impairs 2-APB–induced STIM1 C-terminus mutant (S417G)–Orai1 coupling, although less significantly than mutations at the C-terminal site of Orai1. Consistent with this observation, FRET imaging revealed that there was obviously diminished interaction between Orai1-K85E and STIM1 C-terminus mutant (S417G) (Fig 10B). We also choose the red GECI CMV-R-GECO1.2 to examine the effect of 2-APB on cytosolic calcium in HeLa cells co-transfected with YFP-CT (S417G)-CFP, Orai1-K85E-CFP, and CMV-R-GECO1.2. As shown in Fig 10C and E, despite stores remaining full, coexpression of YFP-CT (S417G)-CFP with Orai1-CFP caused substantial increases

in Ca$^{2+}$ entry, with the attainment of maximal Ca$^{2+}$ peak upon the addition of 50 $\mu M$ 2-APB, followed by rapid inhibition. Instead, coexpression of YFP-CT (S417G)-CFP with Orai1-K85E-CFP appeared to significantly potentiate but not inhibit Ca$^{2+}$ influx (Fig 10D and E). We also found the activation time of SOCE induced by 2-APB in cells coexpressing YFP-CT (S417G)-CFP and Orai1-K85E-CFP is greatly delayed compared with the cells coexpressing YFP-CT (S417G)-CFP and Orai1-WT (Fig 10F).

## Discussion

STIM1 is a type I single-span membrane protein which is located predominantly in the ER membrane (Liou et al, 2005; Roos et al, 2005; Zhang et al, 2005; Grabmayr et al, 2020). Functioning as a finely tuned ER Ca$^{2+}$ sensor, STIM1 can undergo rapid and reversible translocation into close ER–PM junctions to couple with and activate Orai channels in the plasma membrane after store depletion (Soboloff et al, 2012; Prakriya & Lewis, 2015; Grabmayr et al, 2020). To perform these tasks, STIM1 is equipped with several specialized domains spread across its N- and C-terminal portions (Soboloff et al, 2012; Prakriya & Lewis, 2015). Over the past years, a sophisticated

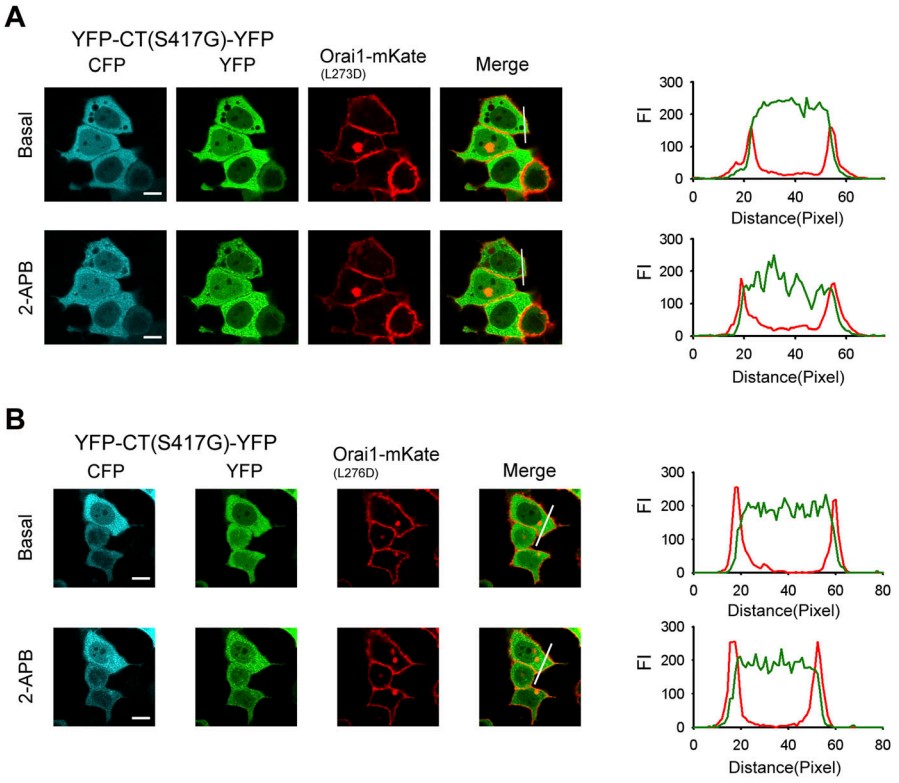

**Figure 8. Orai1 L273D or L276D mutant show disrupted 2-APB–induced association with the STIM1 C-terminus mutant (S417G).**
**(A)** YFP-CT (S417G)-CFP coexpressed with Orai1-L273D-mKate in HeLa cells failed to redistribute from the cytoplasm to the PM in the presence of 50 µM 2-APB. Line-scan intensity plots depicting the distribution of YFP-CT (S417G)-CFP (green line) and Orai1-L273D-mKate (red line) between the cytosol and PM, as indicated by the solid line before (upper panel) and 5 min after (lower panel) application of 50 µM 2-APB. **(B)** 2-APB failed to induce redistribution of YFP-CT (S417G)-CFP and its interaction with Orai1-L276D-mKate. Line-scan intensity plots depicting the redistribution of YFP-CT (S417G)-CFP (green line) and Orai1-L276D-mKate (red line) between the cytosol and PM, as indicated by the solid line before (upper panel) and 5 min after (lower panel) application of 50 µM 2-APB. Bar = 10 µm.

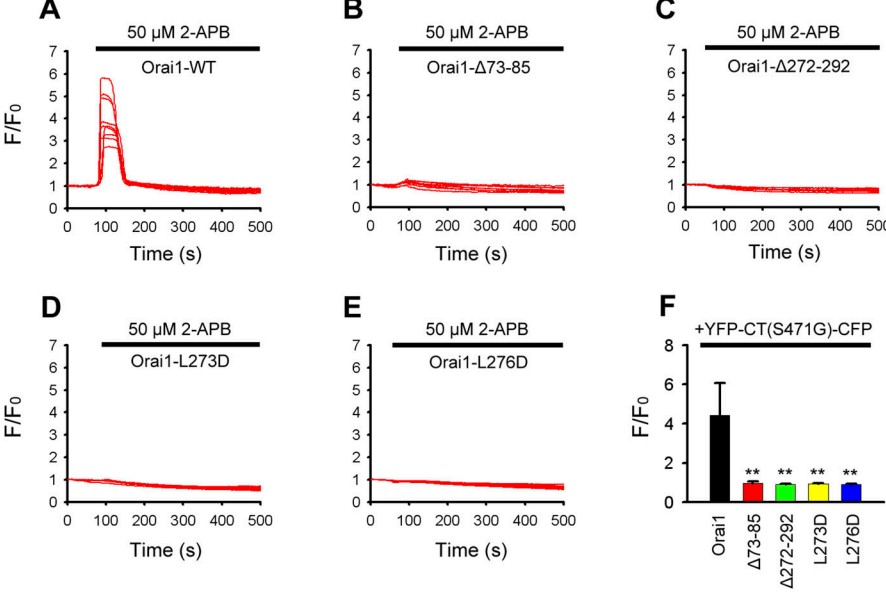

**Figure 9. Both the CBD and NBD of Orai1 are indispensable for 2-APB–mediated STIM1-CT mutant (S417G)–Orai1 binding and Orai1 activation.**
**(A, B, C, D, E)** Representative intracellular free $Ca^{2+}$ traces show 2-APB–triggered $Ca^{2+}$ entry in HeLa cells transfected with (A) YFP-CT (S417G)-CFP+Orai1-mKate; (B) YFP-CT (S417G)-CFP+Orai1-ΔNBD-mKate; (C) YFP-CT (S417G)-CFP +Orai1-ΔCBD-mKate; (D) YFP-CT (S417G)-CFP+Orai1-L273D-mKate and (E) YFP-CT (S417G)-CFP+Orai1- L276D-mKate. **(A, B, C, D, E, F)** Averaged peak $[Ca^{2+}]_i$ values associated with 2-APB–triggered $Ca^{2+}$ influx in cells coexpressing YFP-CT (S417G)-CFP with WT or mutant Orai1 (from left to right, n = 28, 27, 29, 27, and 29) shown in panels (A, B, C, D, E). Graphs show mean ± SD. *$P < 0.05$; **$P < 0.001$.

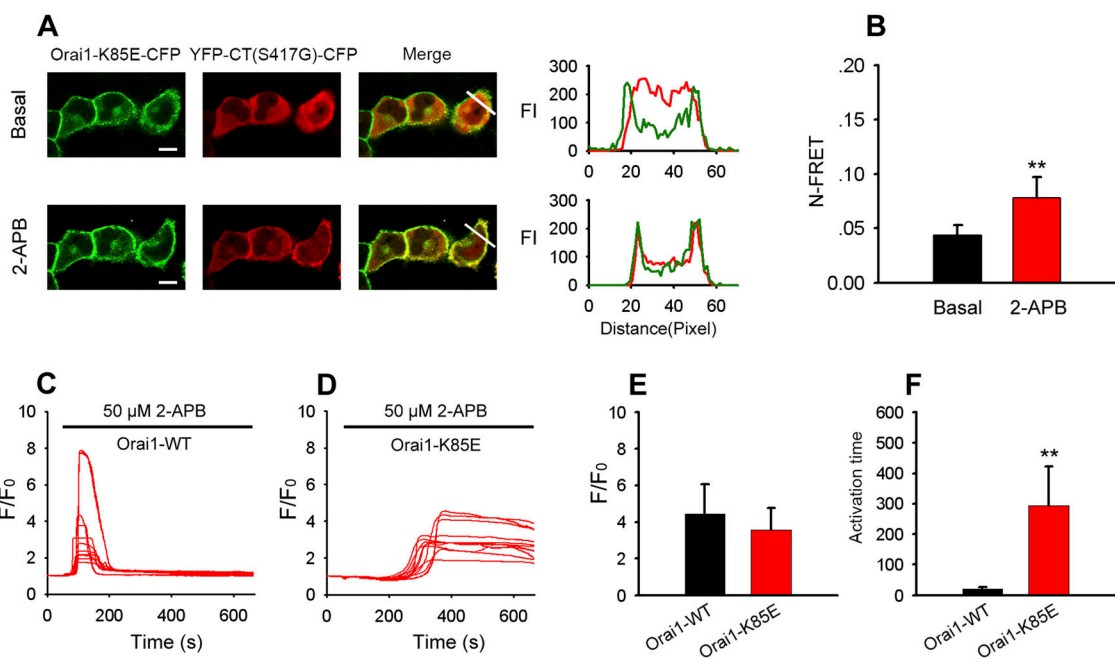

**Figure 10. 2-APB induced STIM1-CT mutant (S417G)–Orai1-K85E binding and Orai1-K85E channel activation.**
**(A)** Representative confocal images of HeLa cells coexpressing YFP-CT (S417G)-CFP with Orai1-K85E-CFP before (upper panel) and 5 min after (lower panel) application of 50 μM 2-APB. Line-scan intensity plots show the distribution of Orai1-K85E-CFP (green lines) and YFP-CT (S417G)-CFP (red line). **(B)** Averaged N-FRET of Orai1-K85E-CFP and YFP-CT (S417G)-CFP expressed in HeLa cells before and 5 min after application of 50 μM 2-APB (from left to right, n = 23 and 22). **(C, D)** 2-APB–induced $Ca^{2+}$ influx in HeLa cells coexpressing YFP-CT (S417G)-CFP and WT-Orai1 (C) or mutant Orai1-K85E (D). **(C, D, E)** Averaged peak $[Ca^{2+}]_i$ values associated with $Ca^{2+}$ influx induced by 2-APB in cells coexpressing YFP-CT (S417G)-CFP and WT-Orai1 (C) or mutant Orai1-K85E (from left to right, n = 28 and 33) shown in panels (C, D). **(C, D, F)** Averaged activation time associated with $Ca^{2+}$ influx induced by 2-APB in cells coexpressing YFP-CT (S417G)-CFP and WT-Orai1 (C) or mutant Orai1-K85E (from left to right, n = 28 and 33) shown in panels (C, D). Graphs show mean ± SD. *$P < 0.05$; **$P < 0.001$. Bar = 10 μm.

model of $Ca^{2+}$-store-depletion–triggered STIM1 activation has been developed, emphasizing the role of both N- and C-terminal segments of STIM1. Recent evidence has provided support for the existence of two "brakes" (the EF-SAM domain itself and CC1 structural inhibitory clamp) on STIM1 activation (Muik et al, 2011; Yang et al, 2012; Zhou et al, 2013; Ma et al, 2015; Prakriya & Lewis, 2015; Rathner et al, 2021). The pioneering studies of Lewis, Prakriya, Zhou, Hogan, Romanin, and other groups have revealed that in cells with replete Ca2+ stores, an inhibitory clamp formed by the CC1 domain interacting with the CC3 domain of STIM1 helps keep STIM1 in an inactive state (Muik et al, 2011; Zhou et al, 2013; Fahrner et al, 2014; Ma et al, 2015; Prakriya & Lewis, 2015; Lunz et al, 2019). $Ca^{2+}$ store depletion activates STIM1 by inducing the intra-dimeric binding of two EF-SAM domains, which triggers conformational changes in the transmembrane (TM) domain, propagates a series of downstream events including release of CAD/SOAR from CC1 inhibition, and the consequent orai1 coupling and activation (Prakriya & Lewis, 2015; Shim et al, 2015). CAD (or SOAR) is recognized as the minimal region required for CRAC channel activation and has been documented to be sufficient to activate SOCs (Kawasaki et al, 2009; Park et al, 2009; Yuan et al, 2009). It contains two putative coiled-coil regions (CC2 and CC3) whose interaction helps to maintain each hairpin monomer in an inactive state (Park et al, 2009; Yang et al, 2012). CC2 has been shown to establish the binding interactions with Orai1, which is a structurally best-defined region in CAD (or SOAR) (Stathopulos et al, 2013). However, Shrestha et al recently found

that in addition to residues in CC3, CC2 and the apex domains of CAD also play essential roles in the maintenance of the inhibitory clamp (van Dorp et al, 2021; Shrestha et al, 2022). Together, the structural details of the CC1–CAD interactions that control release of the intramolecular inhibitory clamp are incompletely understood. Further investigation is needed to reveal it.

S417 is located in the third coiled-coil (CC3) domain of STIM1-CT. In our experiments, multiple kinds of YFP-labeled STIM1 (S417G) mutants of different lengths (containing aa 1–685, 233–685, and 342–488, respectively) were constructed to observe the interaction between STIM1 and Orai1. We found that instead of clear membrane targeting of STIM1 segments exhibited in cells coexpressing Orai1-CFP and STIM1-CT-YFP(aa, 233–685) or STIM1-CAD-YFP (aa, 342–448), either STIM1-CT or STIM1-CAD mutant of S417G led to itself completely localized in the cytoplasm (Figs 1A, C, and D and 3A). When expressed as full-length protein (aa, 1–685), STIM1 (S417G) mutants distributed diffusely in ER in resting cells and remained in ER and failed to aggregate into discrete puncta upon store depletion (Fig 4B). Equally, the coexpressed Orai1-YFP was still homogenously distributed within PM after store depletion, without signs of puncta formation and no significant co-localization with STIM1 (S417G) mutants. STIM1 homomerization is an essential step in the course of STIM1 activation. To further investigate the role of S417 in STIM1 homomerization, we measured FRET between STIM1 proteins in live cells. Our results show that store depletion induced a robust increase in relative FRET between WT STIM1 proteins. By contrast, less

increase in E-FRET was observed in cells coexpressing YFP-STIM1-S417G and CFP-STIM1-S417G, suggesting that mutation of S417G interferes with STIM1 homomerization after store depletion (Fig 4F and G). According to our data, mutation of S417G abolished PM localization of C-terminus or CAD of STIM1, eliminating the FRET between STIM1-CT or CAD and Orai1 and the downstream CAD-mediated $Ca^{2+}$ influx (Figs 3C–F and 4C–E). This indicates that S417 and CC3 may be critical for the binding of CAD to Orai1 and the following activation of CRAC channel. In addition, the S417G OASF and S417G CT sensor mutants showed significant FRET enhancement compared with the OASF WT form, whereas the FRET of S417G CAD sensor mutant is similar to that of the WT sensor (Fig 2B–E), suggesting that mutation of S417G in CC3 of STIM1 may regulate the structure of the coiled-coil clamp involving the CC1 and CC3 domains which is essential in controlling STIM1 activation. But the exact mechanism by which S417 regulates the binding of STIM1 to Orai1 remains unclear, which needs high resolution structural data of the STIM1-CC3 and Orai1 complex to clarify.

2-APB was originally introduced as a membrane-permeant inhibitor of the IP3 receptor (Maruyama et al, 1997). Although it has subsequently been found to affect a variety of ion channels and transport processes, the most reliable and best-studied effect of 2-APB is its ability to affect the activity of CRAC channel (Bootman et al, 2002; Parekh & Putney, 2005). The mechanism of 2-APB action on CRAC channel still remains unclear, but the complex effects elicited by this drug suggests that it may target multiple processes of CRAC activation (Bootman MD et al, 2002; Ali S et al, 2017).

In our study, we were amazed to find that 2-APB restored the binding of the STIM1 C-terminus (S417G) mutant to Orai1 and dose-dependently activated Orai1 channel (Fig 5). However, 2-APB failed to promote the interaction between Orai1-ΔCBD and the STIM1 C-terminus (S417G) mutant (Fig 7). We also found that 2-APB could not cause STIM1-CT (S417G) mutants to redistribute and co-localize with L273D or L276D mutant of Orai1 at the PM (Fig 8). Both L273 and L276 have been recognized as the key residues for the hydrophobic interaction between Orai1 and STIM1 side chains (Navarro-Borelly et al, 2008; Li et al, 2011). Our finding suggested that L273 and L276 was crucial for 2-APB to trigger the STIM1-CT (S417G)–Orai1 coupling. And we showed that Orai1-ΔNBD could also impair the coupling, although its action seemed less significantly than that of Orai1-ΔCBD (Fig 7), but either terminus deletion of Orai1 markedly reduced the 2-APB–triggered $Ca^{2+}$ entry (Fig 9), implying that both C- and N-terminal STIM1 binding sites of Orai1 are essential for STIM1-Orai1 coupling and SOCE. More than that, the decrease of FRET observed in cells transfected with double-labeled STIM1-CT (S417G), indicating a slightly extended conformation of STIM1-CT (Fig S1). Hence, we supposed that 2-APB might initiate an intramolecular transition in STIM1-CT, thereby facilitating the binding with Orai1. Further investigation is needed to disclose the exact molecular mechanisms for 2-APB–induced STIM1-CT (S417G)–Orai1 coupling and Orai1 channel activation.

Several studies have shown that mutation of K85E in extended transmembrane N-terminal (ETON, aa 73–90) of Orai1 disabled the activation of CRAC channel (Lis et al, 2010; Derler et al, 2013; McNally et al, 2013; Wei et al, 2016) despite this only slightly reduced STIM1 binding. But it is still debated whether the binding of STIM1 to Orai1

ETON is required for gating (Yeung et al, 2020). According to our data, the single point mutation of K85E in Orai1 substantially diminished the redistribution of YFP-CT (S417G)-CFP or its co-localization with Orai1-K85E at the cell surface upon 2-APB treatment (Fig 10A). Likewise, FRET imaging revealed obviously diminished interaction between Orai1-K85E and the STIM1-CT mutant (S417G) (Fig 10B). This implies that K85E mutation significantly impairs 2-APB–induced STIM1-CT (S417G)–Orai1 coupling. Furthermore, according to our data, 50 $\mu$M 2-APB induced maximal $Ca^{2+}$ peak followed by rapid inhibition in cells coexpressing YFP-CT (S417G)-CFP and Orai1-WT-CFP (Fig 10C). Interestingly, in cells that coexpressed YFP-CT (S417G)-CFP and Orai1-K85E-CFP, although the activation time of SOCE induced by 2-APB is greatly delayed, this drug still significantly potentiates but does not inhibit $Ca^{2+}$ influx (Fig 10D). These results support the idea that ETON region of Orai1 N-terminal is necessary for STIM1 binding and channel gating and K85 of ETON might mediate 2-APB's inhibition on SOCE activated by STIM1-bound Orai1.

In summary, we show that S417 in CC3 domain of STIM1 plays an essential role for STIM1 function and activation of SOCE. We also present an experiment model of combined STM1 or Orai1 mutants with 2-APB for better understanding of SOCE activation. Our study confirms that both N- and C-termini of Orai1 are involved in channel gating and coupling with STIM1, and K85 of the ETON region is essential for mediating 2-APB's inhibition on SOCE. Thus, our results provide new understanding on Orai1 activation by STIM1 that is important for the in-depth study of molecular structure of the CRAC channel and future drug design.

# Materials and Methods

## Cell culture and transfection

HeLa cells (American Type Culture Collection) were cultured in DMEM containing 10% heat-inactivated fetal bovine serum, 50 U/ml penicillin, and 50 mg/ml streptomycin. Cells were maintained at 37°C in a humidified incubator set at 5% $CO_2$ and were seeded on 30-mm round glass coverslips in a six-well plate. On the following day, cells were transfected with plasmids using Lipofectamine 2000 (Invitrogen) according to the manufacturer's instructions; 6 h later, the medium was replaced with complete DMEM, and cells were cultured overnight. Cells were used for analyses 48 h later.

## Plasmid construction

Human STIM1 (accession number NM_003156) tagged at the N-terminus with pHluorin (pHluorin-STIM1) was a gift from Dr. PY Xu (Institute of Biophysics, Chinese Academy of Sciences), and CMV-R-GECO1.2 was a gift from Robert Campbell (plasmid # 45494; Addgene). The pHluorin was replaced with ECFP or EYFP to generate ECFP-STIM1 or EYFP-STIM1 plasmids. For double-labeled STIM1 constructs, YFP was cloned into the XhoI and HindIII restriction sites of pECFP-N1, and STIM1 fragments were introduced into the EcoRI and BamHI sites (a.a. 233–685, 233–671, 233–474, and 342–448). The fusion proteins Orai1-YFP and Orai1-mKate were generated by

amplifying full-length Orai1 via PCR and cloning the fragments between the EcoRI and BamHI restriction sites of pEYFP-N1 and mKate-N1 vectors, respectively (Clontech). An N-terminal Orai1 deletion mutant (Orai1-ΔN-mKate, Δa.a. 90–301) was cloned by PCR into the mKate-N1 internal EcoRI and BamHI restriction sites. A C-terminal Orai1 deletion mutant (Orai1-ΔC-mKate, Δa.a. 1–272) was similarly constructed. A C-terminally tagged N-terminal–binding domain (NBD) Orai1 deletion mutant (Orai1-ΔNBD, Δa.a. 73–85) was cloned into the EcoRI and BamHI restriction sites of mKate-N1 and pEYFP-N1 expression vectors (Clontech). A CBD Orai1 deletion mutant (Orai1-ΔCBD, Δa.a. 272–292) was prepared as described above. Orai1 and STIM1 point mutants were generated using the QuikChange XL site-directed mutagenesis kit (Stratagene). The integrity of all clones was confirmed by sequence analysis.

### Solutions and chemicals

For confocal imaging experiments, we used standard extracellular Ringer's solution containing the following (in mM): 150 NaCl, 5 KCl, 1.8 $CaCl_2$, 1 $MgCl_2$, 8 glucose, and 10 HEPES (pH 7.4, adjusted with NaOH). $CaCl_2$ was replaced with 1 mM EGTA and 2 mM $MgCl_2$ in $Ca^{2+}$-free Ringer's solution. Stock solutions of thapsigargin and 2-APB were prepared in $Me_2SO$ at a concentration of 1 mM. Fura-2/AM was purchased from Invitrogen. Unless otherwise specified, all reagents and chemicals were from Sigma-Aldrich.

### Confocal microscopy and FRET measurements

The Olympus FV1000 laser scanning confocal microscopy system (Olympus) was used for experiments. Coverslips seeded with HeLa cells transiently transfected with various vectors were placed in a perfusion chamber on the stage of an Olympus IX81 inverted microscope. Images were acquired at room temperature with a 40× or 60× oil objective (N.A. 1.4; Olympus), and images were analyzed with FluoView software (Olympus). CFP, YFP, and mKate/R-GECO1.2 were excited at 405, 514, and 559 nm, respectively.

For FRET experiments, CFP, YFP, and FRET fluorescence was collected with the following parameters: CFP: 405 nm excitation, 450–510 nm emission; YFP: 514 nm excitation, 540–625 nm emission; FRET: 405 nm excitation, 540–625 nm emission. Image acquisition was performed using FV10-ASW Ver.3.1 software. Image analysis was performed with Matlab7.0 software to calculate N-FRET (normalized FRET) according to the equation: $NFRET = (I_{DA} - aI_{AA} - dI_{DD}/I_{DD} \times I_{AA})$, where $I_{DA}$, $I_{DD}$, and $I_{AA}$ are the background-subtracted FRET, CFP, and YFP images, respectively (a = 0.0174 and d = 0.1729). In a second method employed for time-lapse experiments in which dynamic changes in FRET in response to 2-APB stimulation were tracked, a simplified relative FRET approach was used to collect CFP and FRET images, and the ratio of signals obtained in the respective channels ($I_{DA}/I_{DD}$) upon 405-nm excitation was taken as evidence of a FRET change.

### Intracellular $Ca^{2+}$ measurements

Coverslips seeded with HeLa cells transiently co-transfected various vectors with the red GECI CMV-R-GECO1.2 were placed in a perfusion chamber on the stage of an Olympus IX81 inverted microscope. Images were acquired at room temperature with a 40× or 60× oil objective (N.A. 1.4; Olympus), and images were analyzed with FluoView software (Olympus). R-GECO1.2 was excited at 559 nm, and the emission was recorded at 570–670 nm. $Ca^{2+}$ fluctuations are reported as the fluorescence intensity of R-GECO1.2.

### Statistical analysis

Statistical analyses were carried out using the SigmaPlot 11 software (Systat Software Inc.). All of the data indicate the mean ± SD, with sample number (n) referring to cells number. Statistical significance was tested using either $t$ test or analysis of variance. Significance is denoted as $*P < 0.05$ and $**P < 0.01$.

# Supplementary Information

# Acknowledgements

We are grateful to Professor PY Xu at Institute of Biophysics, Chinese Academy of Science, for the kind gift of the EGFP-STIM1 plasmid. We thank Professor YH Liao at Huazhong University of Science and Technology for Orai1-myc constructs. This work was supported by the National Natural Sciences Foundation of China (Grant No. 31371217) and the National Natural Sciences Foundation of China (Grant No. 30871311).

## Author Contributions

T Yu: investigation and writing—original draft.
X Li: investigation.
Q Luo: investigation.
H Liu: investigation.
J Jin: investigation.
S Li: investigation.
J He: resources, data curation, software, formal analysis, supervision, funding acquisition, validation, investigation, methodology, project administration, and writing—original draft, review, and editing.

## Conflict of Interest Statement

The authors declare that they have no conflict of interest.

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
