## [Reviewer comments · Life Science Alliance]

Life Science Alliance

S417 in the CC3 region of STIM1 is critical for STIM1-Orai1 binding and CRAC channel activation

Tao Yu, Xi Li, Qianqian Luo, Huajing Liu, Jing Jin, Shengjie Li and Jun He

DOI: <https://doi.org/10.26508/lsa.202201623>

Corresponding author(s): Dr. Jun He (School of Basic Medical Sciences, Tongji Medical College, Huazhong University of Science and Technology)

Review Timeline:

Submission Date:	2022-07-21
Editorial Decision:	2022-09-12
Revision Received:	2022-12-09
Editorial Decision:	2023-01-03
Revision Received:	2023-01-09
Accepted:	2023-01-09

Scientific Editor: Novella Guidi

Transaction Report:

September 12, 2022

Re: Life Science Alliance manuscript #LSA-2022-01623-T

Dr. Jun He
Tongji Medical College
Huazhong University of Science and Technology
Department of Histology and Embryology
Wuhan, Hubei 430030
China

Dear Dr. He,

Thank you for submitting your manuscript entitled "S417 in the CC3 region of STIM1 is critical for STIM1-Orai1 binding and CRAC channel activation" to Life Science Alliance. The manuscript was assessed by expert reviewers, whose comments are appended to this letter. We invite you to submit a revised manuscript addressing the Reviewer comments.

Thank you for this interesting contribution to Life Science Alliance. We are looking forward to receiving your revised manuscript.

Sincerely,

B. MANUSCRIPT ORGANIZATION AND FORMATTING:

Reviewer #1 (Comments to the Authors (Required)):

The manuscript investigates the impact of loss-of-function STIM1 mutation (S417G), and the authors show that its molecular mechanisms in abolishing SOCE via the inhibition of CAD-Orai1 binding and Orai1 channel activation, the elimination of STIM1 puncta formation and co-localization with Orai1 and SOCE. The authors showed that both the regions of Orai1 (N- and C-termini) are involved in channel gating and coupling with STIM1, with consequent SOCE activation. Moreover, using 2-APB, they demonstrated that its inhibition on SOCE needs the K85 of ETON.

Overall, in the manuscript there are some details that can improve understanding of the work. For example, there is some discrepancy in the reference to the figures: references and details to figures 5C; 7C; 7D; 7E; 7F; 10E and 10F are missing in the text.

Furthermore, it could be more fluid by citing the figures (main and supplementary) in the text sequentially, to make the description and the interpretation of the results of the results obtained more followable and clear.

Finally, some details are missing, such as bars in confocal microscopy images and information related to the magnification of individual images.

Reviewer #3 (Comments to the Authors (Required)):

S417 in the CC3 region of STIM1 is critical for STIM1-Orai1 binding and CRAC channel activation

Tao Yu, Xi Li, Huajing Liu, Jing Jin, Qianqian Luo, Shengjie Li, Jun He

The title describes the function and role of amino acid S417 in STIM1 as critical for STIM1 - Orai1 coupling and thus for CRAC channel activation. Previously known from the literature are the neighboring positions L416, V419 and L423, which lead to gain of function of STIM1 upon loss of hydrophobicity at the respective position. This makes the title sound very exciting and promising. However, the present manuscript is unfortunately incomplete, deficient and partly chaotic as well as confusing due to the use of 2-APB or different Orai1 mutants. Therefore, this paper would need to be restructured and especially completed with control experiments. If the influence of the S417G mutant is to be shown and mechanistically analyzed, I propose the following experiments for this manuscript:

First, the influence of the mutant STIM1 S417G on the CRAC channel system has to be shown (colocalization experiments and calcium imaging experiments), relative to STIM1 wt:

YFP-STIM1 wt + CFP-Orai1 (resting condition)
YFP-STIM1 wt + CFP-Orai1 (thapsigargin (TG))
YFP-STIM1 S417G + CFP-Orai1 (resting condition)
YFP-STIM1 S417G + CFP-Orai1 (thapsigargin)

To propose a coupling defect at this stage would be too speculative. Prior to this, STIM1 homomerization, which is an essential step in the course of STIM1 activation, should be tested (FRET):

YFP-STIM1 wt + CFP-STIM1 wt (-/+TG)
YFP-STIM1 S417G + CFP-STIM1 S417G (-/+TG)

This experiment will show whether the S417G mutation inhibits the homomerization of STIM1 or not.

The homomerization experiments can subsequently also be performed with the STIM1 CT or with OASF or CAD, as these fragments have already been used in the paper. The FRET results are to be presented as a bar diagram.

Since double-labeled STIM1 fragments were already used in the manuscript, this set should also be completed. These "sensor constructs" (STIM1 CT, OASF, CAD) provide information about conformational changes also at the intramolecular level (see PMID: 21427704). FRET results should be shown as bar diagrams:

YFP-STIM1_CT-CFP wt FRET
YFP-STIM1_CT-CFP S417G FRET
YFP-OASF-CFP wt FRET
YFP-OASF-CFP S417G FRET
YFP-CAD-CFP wt FRET
YFP-CAD-CFP S417G FRET

Since only one mutant (S417G) relative to wt is depicted, it would be worth performing the sensor FRET experiments also in the presence of Orai1, since the interaction of OASF and Orai1 is accompanied by a conformational change on OASF (see PMID: 21427704). It would be interesting to see how the S417G mutant performs in this context.

As soon as the manuscript has the complete set of experiments, it should be considered to what extent the 2-APB data are beneficial to understand the effect of S417G.

There are elegant studies and publications about the STIM1 CC3 from the groups like Lewis, Prakriya, Zhou, Hogan. These should be mentioned in the discussion section.

Other remarks:

The manuscript is difficult to read due to many spelling/grammatical errors. I advise a rework of the text in this respect.

What was the rationale for the S417G mutation, i.e., why was glycine ("helix breaker") chosen?

Why did the authors apply conformational sensor constructs normally only used for intramolecular FRET also for intermolecular FRET? This leads to the presence of 2 acceptors for 1 donor (Figure 7) or 2 donors for 1 acceptor (Figure 10). There is no need to complicate the experimental setup with these multiplexed FRET measurements.

On multiple occasions in the manuscript, the authors discuss data that are not shown. These data should instead be included as part of the supplementary information.

The Figures need to be ordered in the sequence that they appear in the text (or vice versa). Additionally, Figure 5C is never mentioned.

The nature of the statistical tests that the authors performed and refer to with P values in most Figure legends remains unclear. It would also be beneficial to add a section to Materials and Methods where the statistical workflow is described.

The legend of Figure 9 does not mention all panels and must be corrected.

Thank you for your letter and for the reviewers' comments concerning our manuscript entitled "S417 in the CC3 region of STIM1 is critical for STIM1-Orai1 binding and CRAC channel activation".(ID: LSA-2022-01623-T). Those comments are all valuable and very helpful for revising and improving our paper, as well as the important guiding significance to our researches. We have studied comments carefully and have made correction which we hope meet with approval. Revised portion are marked in red in the paper. The main corrections in the paper and the responds to the reviewer's comments are as following:

Responds to the reviewer's comments:

Reviewer #1:

1. Response to comment: (Overall, in the manuscript there are some details that can improve understanding of the work. For example, there is some discrepancy in the reference to the figures: references and details to figures 5C; 7C; 7D; 7E; 7F; 10E and 10F are missing in the text.....)

Response: We greatly appreciate the Reviewer's helpful comments. I'm so sorry for our carelessness and we have already changed the relevant references and details to figures in the article. We have also re-written the relevant discussion section and results section.

2. Response to comment: (.....it could be more fluid by citing the figures (main and supplementary) in the text sequentially.....)

Response: According to reviewer's suggestion, we have made correction according to the Reviewer's comments.

3. Response to comment: (.....some details are missing, such as bars in confocal microscopy images and information related to the magnification of individual images.)

Response: We have made correction according to the Reviewer's comments. All confocal microscopy images have been marked with bars.

Special thank to you for your arduous work and instructive advice.

Reviewer #2:

1. Response to comment: (.....To propose a coupling defect at this stage would be too speculative. Prior to this, STIM1 homomerization, which is an essential step in the course of STIM1 activation, should be tested (FRET).....)

Response: We appreciate for the reviewer's valuable comments. According to reviewer's suggestion, we have done relevant experiments to test the effect of S417G mutation on the homomerization of STIM1(Fig4F and G). The results suggest that mutation of S417G interferes with STIM1 homomerization after store depletion. We have also re-written the relevant results section and discussion section.

2. Response to comment: (.....These "sensor constructs" (STIM1 CT, OASF, CAD) provide information about conformational changes also at the intramolecular level.....)

Response: We appreciate for the reviewer's valuable comments. According to reviewer's suggestion, we have done relevant experiments and added relevant data(Fig2B-E). We have also re-written the relevant discussion section and results section.

3. Response to comment: (.....Since only one mutant (S417G) relative to wt is depicted, it would be worth performing the sensor FRET experiments also in the presence of Orail.....)

Response: We should like to thank the reviewer for the helpful comments and have performed relevant experiments. We found that the S417G OASF mutant failed to interact with Orail, suggesting that S417 and the third coiled-coil domain interfered with the extended conformation potentially required for coupling to Orail. In future studies, we will use more biological techniques to explore the relevant mechanisms.

4. Response to comment: (.....The manuscript is difficult to read due to many spelling/grammatical errors. I advise a rework of the text in this respect.....)

Response: According to reviewer's suggestion, we have made correction according to the Reviewer's comments. In future studies, we will use more biological techniques to explore the relevant mechanisms. We tried our best to improve the manuscript and made some changes in the manuscript. These changes will not influence the content and framework of the paper. And here we did not list the changes but marked in red in revised paper.

Special thank to you for your arduous work and instructive advice.

We appreciate for Editors/Reviewers' warm work earnestly, and hope that the correction will meet with approval.

Once again, thank you very much for your comments and suggestions.

January 3, 2023

RE: Life Science Alliance Manuscript #LSA-2022-01623-TR

Dr. Jun He
School of Basic Medical Sciences
Tongji Medical College
Huazhong University of Science and Technology
China

Dear Dr. He,

Thank you for submitting your revised manuscript entitled "S417 in the CC3 region of STIM1 is critical for STIM1-Orai1 binding and CRAC channel activation". We would be happy to publish your paper in Life Science Alliance pending final revisions necessary to meet our formatting guidelines.

- please add ORCID ID for corresponding author-you should have received instructions on how to do so
- please add a Category and Keywords for your manuscript to our system
- please add the Twitter handle of your host institute/organization as well as your own or/and one of the authors in our system
- please consult our manuscript preparation guidelines <https://www.life-science-alliance.org/manuscript-prep> and make sure your manuscript sections are in the correct order
- please use the [10 author names, et al.] format in your references (i.e. limit the author names to the first 10)
- please provide your figure files as single page files (Figure 9 and Figure 10 are currently on the same page)
- please mention Panel E in the Figure 2 legend

A. FINAL FILES:

B. MANUSCRIPT ORGANIZATION AND FORMATTING:

Sincerely,

Reviewer #3 (Comments to the Authors (Required)):

The authors have amended the original manuscript according to most of my suggestions. They have performed some of the suggested experiments, and they provide an important result that reveals that STIM1 S417G shows a defect in homomerization propensity, a prerequisite in the STIM1 activation cascade. The benefit of showing all the experiments with 2-APB is still not fully clear to me, as except of 2-APB partially restoring activity of STIM1 S417G fragments, the additional experiments with the Orai1 mutants do not help much in getting a better understanding of the mechanism behind the S417G mutation.

January 9, 2023

RE: Life Science Alliance Manuscript #LSA-2022-01623-TRR

Dr. Jun He
School of Basic Medical Sciences, Tongji Medical College, Huazhong University of Science and Technology
Department of Histology and Embryology
Department of Histology and Embryology, Tongji Medical College, Huazhong University of Science and Technology
Wuhan, Hubei 430030
China

Dear Dr. He,

Thank you for submitting your Research Article entitled "S417 in the CC3 region of STIM1 is critical for STIM1-Orai1 binding and CRAC channel activation". It is a pleasure to let you know that your manuscript is now accepted for publication in Life Science Alliance. Congratulations on this interesting work.

DISTRIBUTION OF MATERIALS:

Again, congratulations on a very nice paper. I hope you found the review process to be constructive and are pleased with how the manuscript was handled editorially. We look forward to future exciting submissions from your lab.

Sincerely,
